# Potential of Curcumin in the Management of Skin Diseases

**DOI:** 10.3390/ijms25073617

**Published:** 2024-03-23

**Authors:** Kamila Kasprzak-Drozd, Przemysław Niziński, Anna Hawrył, Marek Gancarz, Dominika Hawrył, Weronika Oliwa, Magdalena Pałka, Julia Markowska, Anna Oniszczuk

**Affiliations:** 1Department of Inorganic Chemistry, Medical University of Lublin, Chodźki 4a, 20-093 Lublin, Poland; kamilakasprzakdrozd@umlub.pl (K.K.-D.); anna.hawryl@umlub.pl (A.H.); 2Department of Pharmacology, Medical University of Lublin, Radziwiłłowska 11, 20-080 Lublin, Poland; przemyslaw.nizinski@umlub.pl; 3Institute of Agrophysics, Polish Academy of Sciences, Doświadczalna 4, 20-290 Lublin, Poland; m.gancarz@ipan.lublin.pl; 4Faculty of Production and Power Engineering, University of Agriculture in Krakow, Balicka 116B, 30-149 Krakow, Poland; 5Diagnostics Laboratory, Chodźki 10a, 20-093 Lublin, Poland; dominika.hawryl@diag.pl; 6Science Circle of the Department of Inorganic Chemistry, Medical University of Lublin, Chodźki 4a, 20-093 Lublin, Poland; weronikaoliwa281@gmail.com (W.O.); magda1129@interia.eu (M.P.); juliamarkowska5@gmail.com (J.M.)

**Keywords:** curcumin, skin infections, dermatitis, psoriasis, wound healing, scleroderma, anti-aging, angiogenesis

## Abstract

Curcumin is a polyphenolic molecule derived from the rhizoma of *Curcuma longa* L. This compound has been used for centuries due to its anti-inflammatory, antioxidant, and antimicrobial properties. These make it ideal for preventing and treating skin inflammation, premature skin ageing, psoriasis, and acne. Additionally, it exhibits antiviral, antimutagenic, and antifungal effects. Curcumin provides protection against skin damage caused by prolonged exposure to UVB radiation. It reduces wound healing times and improves collagen deposition. Moreover, it increases fibroblast and vascular density in wounds. This review summarizes the available information on the therapeutic effect of curcumin in treating skin diseases. The results suggest that curcumin may be an inexpensive, well-tolerated, and effective agent for treating skin diseases. However, larger clinical trials are needed to confirm these observations due to limitations in its in vivo use, such as low bioavailability after oral administration and metabolism.

## 1. Introduction

The skin is an important organ, responsible for many activities of the body. It is a protective coating, as well as an excretory and secretory organ, a heat regulator, and a sensory receptor. In addition, it plays an active part in numerous metabolic processes. The skin can externalize the state of health of the body and the activities of its internal organs. Changes in the skin accompany other diseases or arise as a reaction to medication. They can also be a consequence of a contact allergy or bacterial, viral, or fungal infection. Many skin diseases are successfully treated with plant-based products. According to the current literature, plant medicines can generally constitute a safe remedy for skin conditions [1]. Such products can be used internally or externally. One such example of a plant product is curcumin (CUR). Although external (topical) use of curcumin is relatively rare, it can be a valuable agent in the treatment of certain diseases [2].

CUR, demethoxycurcumin, and bisdemethoxycurcumin are collectively referred to as curcuminoids and are present in amounts of more than 3%, 1.4%, and 1.2%, respectively [3]. These yellow compounds occur and are isolated mainly from *Curcuma longa* L. (turmeric) rhizomes. It is a plant belonging to the *Zingiberaceae* family [4]. The curcumin IUPAC name is (1E,6E)-1,7-bis(4-hydroxy-3-methoxyphenyl)hepta-1,6-diene-3,5-dione. The CAS number is 458-37-7 [3]. The chemical formula of curcuminoids is shown in Figure 1 [5].

The curcumin molecule was first isolated from *Curcuma longa* L. in 1815, but it was not until 1910 that Polish chemists determined its spatial structure [6]. CUR consists of two phenyl rings with attached hydroxyl and methoxyl groups. The foregoing functional groups may affect the chemical properties and reactivity of the compound. It is worth mentioning that hydroxyl radical (^•^OH) is one of the most reactive and transient reactive oxygen species (ROS) produced in aerobic organisms [7]. The relevant active compound belongs to the polyphenol group of plant secondary metabolites [8]. CUR under the influence of UV radiation and its environment is rapidly degraded to *trans*-6-(4′-hydroxy-3′-methoxyphenyl)-2,4-dioxo-5-hexanal, vanillin, ferulic acid, and ferulomethane [6]. Nonetheless, CUR is characterized by retentive stability under recommended storage conditions (−20 °C) [3].

Curcumin has a broad spectrum of biological potentialities, of which anti-inflammatory and cardioprotective effects are most often investigated. Figure 2 outlines the therapeutic potential of CUR, specifically focusing on its effects on skin health [3,9], which is a diagram of the following review paper.

The bioavailability of curcumin is scarce. This can be attributed to factors such as low water solubility, low absorption, and rapid metabolism [10,11]. The lipophilic nature of curcumin is a major contributor to its poor absorption [11]. CUR metabolites show the same physiological and pharmacological properties, as well as potency, and are metabolized mainly in the liver and intestines with the participation of intestinal microflora. As a result of biphasic metabolism in the large intestine, the enzymes of the large intestine metabolize the discussed compound, and this occurs in two phases. The transformation also takes place through the interaction of enzymes produced by intestinal microflora in the colon. Intestinal transformations of curcumin involve several stages and different classes of microbial enzymes. For this reason, the composition of the intestinal microflora will result in differences in the biotransformation of curcumin in the diet [10]. The serum concentration of curcumin usually peaks at 1 to 2 h after ingestion and steadily declines within 12 h [12].

To increase the solubility of curcumin, complexes of this compound with albumins or ions of certain metals (e.g., Se^2+^, Zn^2+^) can be used. CUR absorption may also be enhanced by the addition of piperine, nanoparticles, liposomes, phospholipids, or the use of structurally altered curcumin analogs [6]. Formulations (drug delivery systems) of CUR for topical use created to increase bioavailability are for example: chitosan-alginate microcapsules, nano-emulsion [2], or combined with β-cyclodextrin to form a nanoparticle complex [2,13], polymeric bandage [14], methoxy poly(ethylene glycol)-graft-chitosan composite film containing curcumin nanoformulation [15], gel-core hyalurosome (nanovesicle) [16], poly(ε-caprolactone) (PCL) nanofibers [17], liposomal gel [18]. By dint of novel drug delivery systems, curcumin can penetrate the dermis and may constitute an appropriate therapeutic agent for locally treating various inflammatory skin conditions [2]. The molecular mechanisms of curcumin activity in skin diseases are complex and not fully understood yet. In Figure 3, the most probable pathways of CUR action in dermatological issues are shown, yet they are subsequently reviewed in the following sections of this article.

This review summarizes the available information on the therapeutic effect of curcumin in treating skin diseases. As the literature search engine, the browsers in the Scopus, PubMed, Web of Science databases, and ClinicalTrials.gov register were used. The following inquiries were used: “skin diseases and curcumin”, “skin disorders and curcumin”, “psoriasis and curcumin”, “scleroderma and curcumin”, “dermatitis and curcumin”, “wound healing and curcumin”, ”skin infections and curcumin”, “anti-aging and curcumin”, “angiogenesis and curcumin”, “curcumin and ‘in vitro’ studies”, “curcumin and ‘clinical trials’”, “curcumin and ‘animal models’”. Documents published from 1980 to January 2024 were included. In order to qualify work for the review, the following exclusion criteria were applied: works do not contain original data, e.g., reviews, comments, works that have not been independently peer-reviewed, e.g., conference papers, letters to editors, pre-prints, etc., works in languages other than English, papers published before the year 1980. 

## 2. Curcumin and Skin Diseases

### 2.1. Psoriasis

Psoriasis is a chronic, systemic, inflammatory autoimmune disorder. Like other serious skin diseases, visible signs of psoriasis might cause many problems in maintaining proper social relationships. This psychological burden induced by the disease seems to be as important as its physical symptoms, and it can result in impairment of life quality [19]. According to recently published data, the prevalence of psoriasis varies in different geographical regions. The global prevalence rate of psoriasis is about 2–3%; however, the highest risk is seen in some high-income North European countries (up to 11% of the population) [20]. Several clinical types of psoriasis have been reported so far, yet plaque psoriasis (psoriasis vulgaris) is currently the most common form of the disease. About 90% of all reported psoriasis cases are plaque psoriasis [19]. Observed symptoms manifest as erythematous, scaly plaques on the skin. The intensity, localization, and severity of the silvery plaques can vary in patients and may affect any part of the skin (in terms of location); however, the most common sites are the scalp, nails, forearms, and legs [19]. 

Current theories of the pathogenesis of psoriasis are suggesting that immune and inflammatory processes play crucial roles. To date, the interleukin-23/type 17 T cell (IL-23/T17) pathway with related cytokines and chemokines are perceived as the main factors in plaque psoriasis development [21,22]. According to this model, two specific autoantigens, LL-37/cathelicidin and ADAMTS-like protein 5, the synthesis and secretion of which can be directly and indirectly regulated by IL-17, are presented on the surface of dendritic cells (DCs) to T17 cells. Activated T17 cells start producing IL-17, IL-26, IL-29, and TNF-α (tumor necrosis factor-α). IL-17 alone and in synergy with TNF-α induces the transcription of many psoriasis-related genes in keratinocytes [21,23]. It results in the uncontrolled proliferation of these cells and in psoriasis symptoms. A number of medications are currently being investigated or have been approved to target this axis.

The beneficial effects of CUR in the improvement of psoriasis are related to its anti-inflammatory, antioxidant, and immunomodulating properties [9]. It has been demonstrated that CUR is a potent, selective Phosphorylase kinase (PhK) inhibitor. PhK activity is related to activation of the nuclear factor kappa-light-chain-enhancer of activated B cells (NF-κB), which leads to the massive, uncontrolled proliferation of keratinocytes and the development of psoriatic plaque [24]. Indeed, altered levels of PhK have been found in untreated or topical therapeutic treated plaque samples as compared to non-psoriatic skin samples taken from healthy subjects. Reduced levels of PhK in patients treated with 1% alcoholic curcumin gel, as well as in those using other efficient topical agents (i.e., calcipotriol) have also been found when compared to untreated plaque samples [25]. 

In a randomized, double-blind, placebo-controlled, prospective clinical trial in Iran, an improvement of cutaneous symptoms and quality of life in patients with scalp psoriasis has been reported after daily topical administration of turmeric tonic [26]. Similar results have been achieved when a 0.5% CUR microemulgel made of turmeric hydroalcoholic extract was used in the treatment of plaque psoriasis. An excellent safety profile was also highlighted, only mild and relatively rare adverse events were reported during 4 weeks of clinical trialing [27]. 

Inhibition of NF-κB might be also achieved with curcumin administration due to its ability to suppress the excessive production of TNF-α by activated macrophages. CUR can block the TNF-dependent activation of NF-κB using its affinity to the receptor-binding sites of TNF-α [28]. In preclinical studies, topical administration of 1% curcumin gel was found to be a potent anti-inflammatory agent, when applied on psoriasis-like skin alterations in mouse ears induced with imiquimod. The results show decreased levels of IL-17A, IL-17F, IL-22, IL-1β, and TNF-α in the studied group when compared to the control [29]. Similar results were obtained in psoriasis-phenotype transgenic mice after 20 mg/kg of oral administration of CUR. Herein, serum levels of IFN-γ, TNF-α, IL-2, IL-12, IL-22, and IL-23 were significantly lower than in the control group receiving saline. Also, visible symptoms of psoriasis such as ear redness or thickness were significantly reduced after treatment [30].

A new hypothesis of the aetiology of psoriasis has been recently proposed, where progranulin (PGRN), a cell-derived growth factor, might play a key role in inflammatory processes in the course of psoriasis; however, its role is complex and not fully understood yet [31]. It has been also noted that PGRN was significantly upregulated in keratinocytes from psoriatic patients when compared to perilesional and normal skin samples [31,32]. What is more, it was demonstrated that intragastrical administration of CUR in artificially induced psoriatic mice for 21 days in a row suppressed PGRN overexpression in keratinocytes [33]. Interestingly, intragastric administration of CUR in psoriatic mice may not only result in a reduction in pro-inflammatory molecules expression (IL-6, IL-17A, IL-22, IL-23, TNF-α, and TGF-β1), but also in regulation of the gut microbiota. Recent findings suggest that CUR can promote the abundance of selected genera (i.e., *Alistipes*, *Rikenella*, and *Mucispirillum*) in intestinal flora, which were positively correlated with indirect anti-inflammatory activity [34]. 

Curcumin shows poor bioavailability from its classical pharmaceutical forms, oral as well as topical, due to its low solubility and poor skin permeability [35]. Indeed, a low response rate (16.7%) of patients with plaque psoriasis was achieved in a pioneering clinical trial, where 500 mg of encapsulated curcumin was administered three times a day. The authors concluded that one of the possible explanations for these unsatisfactory outcomes was the poor oral bioavailability of curcumin. Moreover, CUR was found to stain the skin yellow—which may be unacceptable to patients [36]. 

A number of novel dosage forms have been recently proposed to overcome this limitation of CUR efficacy in the improvement of psoriasis. Interesting outcomes in topical administration have been reported when CUR was used in the form of nano-emulgel [37], bi-phasic emulgel with tea tree oil [38], microemulsions [27], peptide-modified liposomes loaded with curcumin [39], nanostructured lipid carriers [40], nano-sponges [41], invasosomes [42], and smartPearls^®^ technology [43,44]. Some innovative oral forms of CUR have been also investigated. In a randomized double-blind clinical trial, the efficacy of the lecithin delivery system of curcumin was tested. Patients with mild-to-moderate psoriasis symptoms were randomly divided into two groups, where one received oral CUR (2 g daily) and topical steroids, and the second—only topical therapy. A significant reduction in psoriasis symptoms was achieved in both groups, yet better results were obtained in the group receiving both CUR and steroids than steroids alone. What is more, in the CUR group, IL-22 serum levels were significantly decreased [45]. 

Nanoparticles can be also delivered via the oral route of administration. Promising results have also been reported in a placebo controlled, double-blind randomized clinical trial, where one group of patients with moderate-to-severe psoriasis was treated with acitretin (a second-generation retinoid) with nanoparticles of curcumin, and the other group was treated with only acitretin for 12 weeks. A significantly greater reduction in psoriasis symptoms was observed in patients receiving CUR. Notably, patients in the control group had significantly higher total serum cholesterol levels; therefore, curcumin potentially plays a protective role against acitretin-related dyslipidemia [46]. All of these innovative pharmaceutical forms show excellent safety profiles and efficacy in vitro and in vivo, which make them promising candidates for new, more efficient therapeutic approaches. 

### 2.2. Scleroderma

Scleroderma or systemic sclerosis (SSc) is a chronic, immune-mediated rheumatic disease, resulting in fibrosis and vasculopathy of skin and internal organs. SSc also may affect the GI tract, lungs, kidneys, heart, and musculoskeletal system; it can also induce pulmonary hypertension [47]. Fibrosis in the course of SSc is caused by the increased production and deposition of extracellular matrix (ECM) components in different tissues and organs [48]. Although SSc is a relatively rare disease, affecting less than two per 100,000 people annually [49], it is characterized by high morbidity and mortality, often caused by a delay in the ultimate diagnosis and implementation of appropriate treatment [48]. Systemic sclerosis can be divided into two subsets that vary in the spectrum of skin involvement—limited systemic sclerosis (lcSSc) and diffuse systemic sclerosis (dsSSc). In lcSSc, progression of the disease is relatively slow when compared to dsSSc, and it is reflected in the prognosis—10 year survival is estimated for 90% and 65% to 82%, respectively [47]. 

CUR has been found to be an effective anti-fibrotic agent, and may have a beneficial effect on maintaining balance in the TGF-β/Smad signaling pathway. In the study by Hsu et al., it was reported that CUR can reduce the expression of ECM proteins in the keloid fibroblasts via inhibition of the TGF-β1/Smad pathway in a dose-dependent manner [50]. Ryu et al. suggest that CUR may decrease type I collagen expression in SSc, hypertrophic scars, and keloids through inhibiting the expression of Smad2/3, and increasing the expression of MMP-1, a type I collagen degrading enzyme in human skin fibroblasts [51].

Another study on human fibroblasts cell cultures resulted in similar findings and also suggests that CUR can upregulate TGF-β induced factor (TGIF)—a known TGF-β suppressing factor [52]. The derived data revealed that CUR-mediated suppression of the TGF-β1/Smad pathway in the course of SSc and other fibrotic diseases is complex and requires further studies. 

The beneficial role of curcumin in scleroderma is also related with its ability to modulate protein kinase C (PKC) signaling, which has been identified as another important pathway in the development of tissue fibrosis. Jimenez et al. reported that PKC isoform δ (PKCδ) may play a crucial role in the upregulation of type I collagen expression in human fibroblasts cultures derived from skin samples of the patients with Ssc [53], whereas Conboy et al., and thereafter Wermuth et al., found CUR as a potent PKCδ inhibitor in vitro [54,55]. 

Interestingly, in scleroderma lung fibroblasts (SLF), CUR induces apoptosis; however, this does not occur in normal lung fibroblasts (NLF). The effect may be due to the lower level of PKC isoform ε (PKCε) in SLF and failure of the upregulation of phase 2 detoxification enzymes in response to CUR. A deficit of glutathione-S-transferase (GST) is a direct consequence of PKCε alteration in SLF; however, the upregulation failure of heme oxygenase-1 (HO-1) is independent from PKCε expression [56,57]. 

Despite the fact that a number of in vitro and in vivo studies show promising results and indicate CUR as a potentially efficient agent in the treatment of scleroderma, we found no ongoing or closed clinical trials regarding CUR efficacy in SSc. Further studies on the molecular mechanisms of curcumin action in the development and progression of SSc are needed in order to better understand its potential and future perspectives in clinical trials.

### 2.3. Atopic Dermatitis

Atopic dermatitis is a chronic inflammatory skin disease that primarily affects children [58]. It is caused by dysregulation of the immune system and damage to the skin barrier [59]. Allergens act on the damaged skin, causing itching and subsequent scratching, which results in eczematous skin lesions. The pathogenesis of atopic dermatitis is characterized by an imbalance in the different T-lymphocyte subgroups. In the early stages of the disease, Th2 cells produce cytokines such as IL-4, IL-5, IL-13, and IL-31 in increased concentrations. In the later stages, there is a shift from an initial Th2 response to a Th1-type immune response. This is characterized by the excessive release of IL-1, IL-6, IL-12, IL-18, and TNF-α by the recruited monocytes [60]. The presence of inflammatory cells results in oxidative stress, which exacerbates the inflammatory response in the skin [58]. The onset of the disease in early childhood can often lead to the development of allergic rhinitis and asthma, a phenomenon known as the atopic march [61].

Curcumin has been used to treat symptoms of atopy and pruritic skin since the time of Ayurveda. In a study on a human mast cell line, Moon et al. [62] reported a significant reduction in the expression of thymic stromal lymphopoietin, a key factor in atopic dermatitis, when treated with CUR. Sharma et al. [58] evaluated, in turn, the potential of CUR in alleviating ovalbumin-induced atopic dermatitis in mice. In their study, female BALB/c mice were treated with an egg albumin patch applied to the skin for 1 week. After a two-week period without albumin application, the same protocol was repeated three times. During the last phase of albumin exposure; CUR was administered daily at a dose of 20 mg/kg (i.p.) for seven consecutive days. This secondary plant metabolite alleviated the ovalbumin-induced skin disease, as evidenced by the normalization of epidermal thickness and the inhibition of inflammatory cell infiltration in the dermis. The results indicate that the administration of curcumin can reduce the development of atopic dermatitis in experimental animals.

Rawal et al. [63] evaluated the effectiveness and safety of Herbavate^®^ cream, a multi-herb preparation containing CUR, in treating eczema patients on an outpatient basis. Patients (150) applied Herbavate^®^ topically twice daily for four weeks. Changes in eczema symptoms, including itching, erythema, thickening, and flaking, were assessed every seven days. A clinical improvement after four weeks compared to the baseline was significant for all assessed symptoms. The cream was well-tolerated, with only five patients reporting mild adverse effects. However, the study design raises questions about the interpretation of the results due to the lack of a control group, a high percentage of dropouts, and the inability to distinguish the effects of CUR from other cream components. To establish the potential role of CUR in the treatment of atopic dermatitis, further comparative, randomized, clinical studies are needed. Moreover, researchers have shown that preparations containing CUR (such as creams and ointments) may, however, cause contact urticaria and dermatitis [64,65,66]. Therefore, further experiments are needed to investigate the role of curcumin in treating atopic dermatitis.

### 2.4. Iatrogenic Dermatitis

Iatrogenic dermatitis comprises a variety of inflammatory skin conditions that result directly from medications or medical procedures. These conditions include allergic contact dermatitis and radiation-induced dermatitis [9]. As of the time of writing this review, there has been limited preclinical and clinical research conducted on the use of CUR as a natural, safe, and cost-effective treatment for iatrogenic dermatitis. This compound has, however, been found to protect the skin from epidermal damage caused by radiation and to prevent morphological changes resulting from skin irradiation. Therefore, it should help to maintain the proper thickness of the epidermis and the density of cells in the basal layers [67]. 

In a study by Kim et al. [68], the effect of topical curcumin treatment on radiation burns was evaluated using a minipig model. Histological and clinical changes were observed five weeks after exposure to radiation (gamma radiation). Curcumin was topically applied to the irradiated skin twice daily for 35 days at a dose of 200 mg/cm^2^. The results showed that CUR treatment reduced epithelial shedding following irradiation. Furthermore, in related work, patients treated with curcumin exhibited a reduced expression of cyclooxygenase-2 and nuclear factor kappa-B (NF-κB)compared to the control group. In addition, CUR treatment stimulated wound healing. These findings indicate that CUR could be utilized in the management of radiation burns due to its ability to stimulate the regeneration of epithelial cells in the skin and promote wound healing.

Ryan et al. [69] conducted a randomized, double-blind clinical trial to evaluate the potential of CUR in reducing the severity of radiation dermatitis in 30 breast cancer patients who received radiation therapy without concurrent chemotherapy. The patients were administered 2.0 g of CUR (first group) or placebo (second group) orally three times daily (i.e., 6.0 g per day) throughout the course of their radiation therapy. The study found that CUR reduced the severity of radiation dermatitis compared to the placebo. What is more, there were no significant differences observed due to demographics, skin radiation dose, redness, or pain. In summary, patients who were administered with 6.0 g of curcumin orally daily during radiotherapy showed a reduction in the severity of radiation dermatitis.

Pruritus is a chronic skin disease that can significantly reduce quality of life. Panahi et al. [70] investigated the effectiveness of curcumin in alleviating the symptoms of chronic itching caused by sulfur mustard. The study involved 96 patients who were randomly assigned to one of two groups. The first group (n = 46) received a daily dose of 1 g of curcumin for 4 weeks, while the second group (n = 50) received a placebo for the same duration. At the beginning and end of the study, the concentration of serum substance P and antioxidant enzyme activity were measured. In the group taking CUR, there was a significant decrease in the concentration of substance P in the serum and the activity of glutathione peroxidase and catalase as well as superoxide dismutase, while no significant change was observed in the placebo group. Additionally, CUR supplementation was associated with a reduction in itch severity. The CUR group showed also a significantly greater improvement in their quality of life. In summary, CUR may be a natural and safe treatment for chronic itch, but further extensive research on a larger patient population is necessary to confirm its effectiveness.

The aim of subsequent studies was to evaluate whether curcumin is effective in preventing capecitabine-induced hand-plantar syndrome. The study included patients who were naïve to capecitabine and were to receive this drug. The patients (n = 40) received curcumin at a dose of 4 g/day (2 tablets 12 h apart) starting at the beginning of capecitabine treatment and continuing for six weeks. The results showed that CUR combined with capecitabine reduced the incidence of hand–foot syndrome. However, further extensive clinical trials are required to confirm the effectiveness of this compound [71].

### 2.5. Wound Healing

Generally, wounds can be classified as acute or chronic depending on their etiology. Acute wounds are formed as a result of external skin damage and may be surgical wounds, burns, cuts, as well as post-traumatic wounds (lacerations or crushing wounds). Acute wounds usually heal within a predictable time, depending, of course, on the type, location and depth of the wound. Chronic wounds are most often leg ulcers, foot ulcers, and pressure sores, which are most often the result of metabolic diseases (diabetes), obesity, old age, and improper diet. Chronic types of wounds heal slowly and unpredictably [72].

The wound healing process consists of several stages of molecular and cellular events that occur immediately after tissue damage and whose aim it is to rebuild it. The subsequent phases of tissue reconstruction are hemostasis, inflammation, proliferation, and remodeling [73]. The process of hemostasis involves platelet aggregation (formation of blood clots), which provides a temporary extracellular matrix for cell migration [74]. The first event that occurs immediately after the injury is the exudation stage, which contributes to increased tissue swelling. In the next stage—proliferation, the area of tissue damage is reduced by shrinking myofibroblasts and fibroplasia; this stage lasts from 2 to 14 days after the damage. In the next stage, the third stage of wound healing, remodeling occurs to achieve the maximum tensile strength through reorganization, degradation, and resynthesis of the extracellular matrix. The normal structure of the tissue is slowly restored, while the granulation tissue is rebuilt into scar tissue, which is less vascular. The final stage of wound healing lasts from three weeks to a year or even longer [75].

Topical application of curcumin in the form of various preparations such as hydrogels [76], nanoemulsions [77], collagen films [78], polymer bandages [79], or nanofibrous membranes [80] have proven to be very useful for the local treatment of damaged skin.

The anti-inflammatory effect of CUR is demonstrated by regulating inflammatory signaling pathways and inhibiting the production of inflammatory mediators [81]. Detailed in vitro and in vivo studies confirmed that CUR reduces the release of inflammatory cytokines from monocytes and macrophages (interleukin-8, IL-8 and tumor TNF-α) and also inhibits the enzymes involved in inflammation—cyclooxygenase 2 (COX-2) and lipoxygenase (LOX) [17]. Additionally, curcumin reduces angiotensin II-induced inflammatory responses by inhibiting vascular smooth muscle cell proliferation by increasing peroxisome proliferator-activated receptor gamma (PPAR-γ) activity [82]. Another mechanism by which CUR reduces inflammation is to compete with LPS for MD2 binding, thereby effectively inhibiting the TLR4-MD2 signaling complex [83]. Moreover, inflammasome 3 (NLRP3) with the NOD-like receptor containing a pyrin domain are cytosolic multi-protein complexes that are involved in the development of various inflammatory diseases. The inflammasome complex, 3 NLRP3, consists of three components: a sensor protein, an apoptosis-related speckle protein containing a caspase recruitment domain, and a caspase-1 protease. Research has shown that CUR can directly inhibit the accumulation of the NLRP3 inflammasome or inhibit the activation of the NLRP3 inflammasome by inhibiting the NF-κB pathway. This may be one of the mechanisms of CUR in the treatment of inflammatory diseases [81]. 

Reactive oxygen species (ROS) are used as protection against microorganisms present in wounds. However, long-term exposure to high concentrations of ROS causes oxidative stress, which damages cells and is often the cause of wound inflammation. Antioxidant compounds can improve wound healing and scavenge free radicals when they are applied topically [84]. 

CUR also has antioxidant properties, which is confirmed by a number of clinical studies [85,86,87]. The curcumin-incorporated collagen matrix (CICM) was tested for antioxidant activity using the lipid peroxidation method, and the results obtained confirmed its free radical scavenging activity [85]. As a result of long-term inflammation and increased oxidative stress, wound healing in diabetic patients may be difficult. Therefore, studies have also been conducted on the antioxidant activity of CUR in wound healing in diabetic rats. The study results may be an important strategy to improve healing in diabetic patients. The use of CUR increased wound contraction and reduced the expression of inflammatory cytokines/enzymes such as tumor necrosis factor alpha, interleukin (IL)-1beta, and matrix metalloproteinase-9. 

CUR also increases the level of anti-inflammatory cytokines, i.e., IL-10, and antioxidant enzymes, i.e., superoxide dismutase, catalase, and glutathione peroxidase. It has been confirmed that the anti-inflammatory and antioxidant effects of CUR cause faster and better wound healing in diabetic rats [86]. Another study confirmed the antioxidant properties of CUR used in a new wound healing dressing. Here, CUR was added to the cyclic β-cyclodextrin to form the inclusion complex and then integrated into the chitosan-alginate composite mixture. Wounds treated with the CUR complex showed an accelerated rate of closure, improved histopathological results and lower levels of superoxide dismutase and lipid peroxidation. The accelerated wound healing process can be attributed to the antioxidant, anti-inflammatory properties of CUR contained in the complex [87].

In the next phase of wound healing (proliferative), proliferative fibroblasts migrate to the dermis and produce immature ECM proteins (EDA ibronectin and type III collagen) and activating growth factors (TGF-β1) [88]. The process of fibroblast infiltration into the wound site is essential for the formation of granulation tissue, as well as for the production and deposition of collagen [89]. Skin wounds that do not heal within the expected time frame have impaired fibroblast proliferation and migration within the wound. Research has also been conducted to improve the quality of wound healing by slowly delivering curcumin from collagen, which also acts as a supportive matrix [85]. The obtained biochemical parameters and histological analysis showed increased wound reduction, enhanced cell proliferation and effective free radical scavenging in the group treated with CICM curcumin-incorporated collagen matrix.

Other studies have used CUR in a gel form topically on wounds. This form of treatment resulted in rapid closure of the wound with well-formed granulation tissue dominated by collagen deposition and regenerating epithelium. The biological activity of curcumin consisted of increasing the mRNA and protein levels of tumor necrosis factor alpha in the early phase of healing, and the level of collagen was significantly higher in wounds treated with CUR. In summary, topically applied CUR accelerated wound healing in mice by regulating the levels of various cytokines [90]. Another valuable solution for the use of curcumin in wound healing was the preparation of a novel sandwich nanofibrous membrane (CSNM) filled with CUR. In a rat dorsal skin injury model, it was confirmed that CSNM clearly induced the growth of granulation tissue, collagen deposition, and epithelial tissue remodeling [91].

An attempt was also made to confirm that CUR can regulate gene expression in human gingival fibroblasts (hGF). For this purpose, the effect of curcumin on the expression of wound healing-related genes, type I collagen (COL1), keratinocyte growth factor (KGF)-1, and the epidermal growth factor receptor (EGFR) was investigated in an in vitro hGF wound healing model, as well as the hGF pathway signaling involved in the regulation of these genes by curcumin. The study confirmed that CUR induced the expression of KGF-1 and EGFR in intact hGFs. In an in vitro wound healing model, curcumin also increased the expression of COL1 and KGF-1 [92].

### 2.6. Skin Infections

#### 2.6.1. Antibacterial Activity

The pathogens responsible for skin infections include bacteria that naturally occur on the skin as commensals, including microorganisms of the genera *Corynebacteria*, *Propionibacteria,* and *Staphylococci*. However, most skin infections are often caused by *Staphylococcus* spp. especially *Staphylococcus aureus*, which is responsible for infections such as boils, impetigo, cellulitis, and folliculitis. One of the mechanisms of curcumin’s antibacterial action is the blockage of bacterial growth through its structural properties and the production of antioxidant products. Moreover, CUR can inhibit bacterial virulence factors or bacterial biofilm formation through the bacterial quorum sensing regulatory system. CUR also has another interesting property, namely it acts as a photosensitizer and, under the influence of blue light irradiation, causes phototoxicity, inhibiting the growth of bacteria [93]. 

Commonly used treatments for bacterial skin infections are based on incision and drainage and antibiotic therapy. Isolated strains of bacteria responsible for skin infections are often antibiotic resistant, so selecting the right antibiotic or group of antibiotics is difficult. In addition, excessive and incorrect use of antibiotics when treating skin infections and poor infection prevention result in the development of resistant strains of bacteria [94]. 

Antimicrobial photodynamic therapy (aPDT), which creates reactive oxygen species (ROS) leading to the death of bacterial cells, has turned out to be an alternative method for treating resistant *Staphylococcus* strains. One study evaluated the effectiveness of antimicrobial photodynamic therapy (aPDT) CUR in combination with artificial skin for disinfecting an infected skin wound in rats. Research has confirmed the positive benefits of using aPDT with blue light and curcumin in artificial skin to disinfect and accelerate wound contraction [94]. 

Among such resistant microorganisms is methicillin-resistant *Staphylococcus aureus* (MRSA). In vitro studies of the antimicrobial effect of PDT with CUR against MRSA biofilm were performed. The formed MRSA biofilm was incubated with curcumin, then illuminated with an LED (light-emitting diode), and finally, colony-forming units were counted using scanning electron microscopy (SEM). The positive effect of the study (PDT with CUR reduces the growth of MRSA biofilm) may be an alternative to the inactivation of this resistant strain [95].

Other authors have attempted to add a photosensitizer to detergent, which can prevent the development of infection and disinfect surfaces when they are illuminated with light of the appropriate wavelength. A detergent containing a photosensitizing molecule (CUR) was developed and characterized using UV–vis spectroscopy and fluorescence spectroscopy. The antibacterial photodynamic effect of the detergent with CUR was assessed against *Staphylococcus aureus* in a planktonic medium and in vivo (skin infection model). Developed detergents containing curcumin can improve the disinfection of material surfaces and skin infections (in vivo) under the influence of light [96].

CUR, as a hydrophobic compound, is poorly soluble and has cytotoxic effects in large doses. Therefore, another publication attempted to use curcumin-loaded graphene oxide flakes as an effective antibacterial system against MRSA. Graphene oxide is a nanomaterial with a large surface area and also has antibacterial properties, mainly due to the mechanical cutting of bacterial membranes. The ability of graphene oxide to support and stabilize curcumin molecules in an aqueous environment was confirmed, as well as the effectiveness of the graphene oxide–CUR complex against MRSA at concentrations below 2 µg mL^−1^. The complex showed low toxicity towards fibroblast cells and allowed it to avoid hemolysis of its red blood cells [97]. 

The activity of the photosensitive compound curcumin in the presence of a diode laser (405 nm) against a Vm-resistant *S. aureus* biofilm (VRSA) was also confirmed in vitro and in vivo. In this study, we found that intracellular ROS accumulation led to bacterial death without causing toxicity to human cells [98]. Similarly, other authors investigated the effect of combining CUR as a photosensitive substance against MRSA in a mouse model of intradermal infection. Mice were infected intradermally with 108 CFU of *S. aureus*, and then CUR was administered at the site of infection and left in the dark for 30 min, and finally it was treated with LED light (450 nm, 10 min, 54 J/cm^2^). A reduced bacterial load in lymph nodes, as well as hyperplasia of these organs, was observed—compared to control groups [99]. 

One helpful solution may also be the use of CUR in combination with antibiotic therapy. Studies have been conducted to evaluate the antibacterial effect of curcumin in combination with tetracycline or ciprofloxacin against *Staphylococcus aureus* (*S. aureus*) and *Escherichia coli* (*E. coli*). The antibacterial activity of the selected combination was determined in situ based on the recovery rate and wound assessment on rabbit skin infected with *S. aureus* after CUR and antibiotic combination treatment. Imaging was then performed using scanning (SEM) and transmission electron microscopy (TEM) to determine the morphological changes of *S. aureus* treated with CUR combinations. Based on the obtained research, it was confirmed that the combination of curcumin with tetracycline showed a synergistic effect against *S. aureus* and *E. coli*, while the combination of curcumin only with ciprofloxacin showed a synergistic interaction with *S. aureus*. The combination of CUR and tetracycline was selected for further testing, and it was found that the ointment based on the combination of CUR and tetracycline did not cause skin or eye irritation. Furthermore, applying a combination of 2% curcumin with 1% tetracycline to the skin of a rabbit infected with *S. aureus* healed the infected skin faster than 3% tetracycline ointment alone [100].

The antibacterial activity of CUR encapsulated in Carbopol and lipid nanocarriers containing CUR nanostructures (CURC-NLC) were tested against the following bacterial strains (*Bacillus subtilis* ATCC 6633, *Escherichia coli* ATCC 25922, *Salmonella* spp., *Staphylococcus aureus* ATCC 25923, *Staphylococcus epidermidis*). The CURC-NLC preparation showed a strong inhibitory effect on Gram-positive and Gram-negative bacteria, twice as strong as CUR, and its greater potential to accelerate wound healing was observed [101].

Evaluation of curcumin’s antimicrobial activity against MRSA has also been evaluated in diabetic wound infections. It has also been reported that HAMLET (human α-lactalbumin causing cancer cell death) can sensitize bacterial pathogens to traditional antimicrobials. Therefore, the activity of CUR nanoparticles in the healing of diabetic wounds infected with HAMLET-sensitized MRSA was determined. Microbiological examination, planimetric, biochemical, histological and quantitative morphometric tests; immunohistochemical staining, hydroxyproline level determination and reverse transcription polymerase chain reaction for caspase 3 showed a significant difference between animals in the MRSA/CNP/HAMLET group compared to other groups. Thus, it was confirmed that curcumin nanoparticles healed MRSA-infected diabetic wounds sensitized using HAMLET and may represent a safer topical agent for infected diabetic wounds [102].

#### 2.6.2. Antifungal Activity

Similar to antibiotic resistance to a number of bacteria that cause skin diseases, skin infections caused by fungi are often difficult to treat due to their drug resistance. Therefore, new drugs with antifungal or synergistic effects with antifungals are still being searched for [103]. The most common fungal infections are dermatophytoses (dermatomycosis, mycosis, ringworm), i.e., diseases caused by keratinophilic fungi called dermatophytes, which may include skin, hair, and nail diseases. One study examined the antifungal activity of turmeric oil and CUR from *Curcuma longa* L. against fifteen isolates of dermatophytes (pathogenic molds and yeasts). It was confirmed that turmeric oil in appropriate dilutions could inhibit the activity of all dermatophyte isolates, while curcumin did not inhibit any of them [104]. 

The in vitro sensitivity of several *Aspergillus* and *Candida* isolates to CUR encapsulated in hyaluronan nanovesicles was also tested. The studies confirmed that CUR coencapsulated in nanovesicles without hyaluronan had a stronger effect against *Candida* isolates than fluconazole. Examination of skin permeation profiles using an in vitro Franz diffusion cell system showed that CUR was released from the nanobubbles and accumulated in the skin but did not penetrate the skin layers [105]. 

Moreover, the effect of photodynamic CUR therapy on dermatophytes was investigated using an in vitro assay. A well test was performed for the activity of CUR against the conidia of *Trichophyton rubrum*, *Trichophyton interdigitale*, *Trichophyton terrestre*, *Microsporum canis*, *Microsporum gypseum,* and *Epidermophyton floccosum*. The obtained results confirmed that all dermatophytes were significantly inhibited depending on the concentration of CUR, which may encourage further development of photodynamic therapy of mycosis using curcumin [106]. 

Similar results were obtained in another work, where the photochemical inhibition of *Trichophyton rubrum* by various CUR compounds were tested. CUR dissolved in DMSO and irradiation had a clear dose-dependent inhibitory effect on fungal growth, while the same procedure with micellar curcumin had no inhibitory effect [107].

In addition, CUR has found antifungal use in combination with fluconazole in the form of a topical nanoemulsion. A nanoemulsion was optimized in which the globule size was smaller than 200 nm and showed increased permeability through the skin and better antifungal effectiveness compared to the native form of fluconazole and CUR suspension. An antifungal test confirmed the synergistic effect of the combination of fluconazole and CUR against multidrug resistance of *Trychophytum rubrum* and *Trichophyton* metagrophytes compared to fluconazole alone [108]. The aim of the next study was to prepare biocurcumin (CMN) nanoemulsion (CMN-NE) for transdermal administration in the treatment of mycoses. It was shown that the prepared CMN-NE improved bioavailability, with better skin penetration in patients with mycosis [109]. 

The effectiveness of local CUR was also compared with that of nystatin in the treatment of oral candidiasis. A clinical and microbiological evaluation confirmed that synthesized CUR nanoparticles significantly reduce the number of *Candida* colonies. However, no statistically significant difference was observed between groups of infected mice receiving nanocurcumin and nystatin. Due to the increased bioavailability of curcumin at the nanoscale, it can be qualified as an alternative therapy for oral candidiasis, and it also helps to avoid nystatin-related diseases [110].

#### 2.6.3. Antiparasites Activity

One of the skin diseases caused by the parasite is leishmaniasis. In the treatment of this disease, pentavalent antimonias and amphotericin B, miltefosine and paromomycin can be used, which unfortunately have high toxicity indexes. Studies on the use of curcumin in the treatment of leishmaniasis were conducted, where the mechanism of action of ivermectin and CUR on *Leishmania* (*L.*) *amazonensis* promastigotes was compared using electron paramagnetic resonance (EPR) spectroscopy. Based on the results obtained, it was concluded that treating parasites with both compounds causes stiffness of the cell membrane as a result of oxidative processes. Other measurements indicated that ivermectin had a greater affinity for the parasite membrane than CUR, resulting in lower IC50 values in low cell concentration assays. However, studies of higher cell concentration compounds showed no significant differences [111]. 

Another type of therapy for cutaneous leishmaniasis is photodynamic therapy (PDT) using CUR. In the experiments performed, the response of macrophages infected with *L. braziliensis* and *L. major* to PDT using curcumin was assessed. The lowest analyzed concentrations of CUR (15.6 and 7.8 g/mL) showed photodynamic inactivation. Both cell destruction and internalization of CUR in macrophages and intracellular parasites were observed using microscopic techniques. The study results also confirmed an increase in the polarity of the mitochondrial membrane and a decrease in the number of recovered parasites [112]. 

Another methodology for research on the leishmanicidal activity of CUR was carried out in in vitro and in vivo tests. The synthesis of curcumin-coated gold nanoparticles (Cur@AuNPs) was performed using a simple green chemistry technique, and then their stability, cytotoxicity, and leishmanicidal activity were investigated. In vitro leishmanicidal activity against extracellular promastigotes and intracellular amastigotes of the protozoan parasite *Leishmania major* (*L. major*) was performed using a tetrazole reduction colorimetric method. However, in the in vivo test, the size of footpad lesions and the parasite load were measured in two organs of the mouse infection site. The immune responses of the mice were determined by measuring interferon gamma (IFN-γ) and interleukin-4 (IL-4) levels. As a result of the conducted research, the viability of *Leishmania* promastigotes decreased significantly, along with the inhibition of promastigote growth. Application of the nanoparticle in vitro effectively removed *L. major* amastigotes explanted in macrophages and also had no harmful toxic effects on mouse cells [113].

In another publication, CUR nanopreparations were tested for antiacanthamoebic properties. The newly synthesized curcumin-loaded nanovesicles were characterized using Fourier transform infrared spectroscopy, ultraviolet–visible spectrophotometry, and atomic force microscopy. CUR was tested using an amoebicidal test against *A. castellanii* (genotype T4). However, the effect of curcumin nanopreparations on host cells was determined by performing cytotoxicity tests using human keratinocyte cells (HaCat). The obtained results confirmed that CUR in the form of nanobubbles enhanced the antiacanthamoebic effect without damaging human cells, as compared to curcumin [114]. 

The drug of choice for combating trichomoniasis, caused by the protozoan *Trichomonas pochwylis*, is metronidazole. However, increased resistance to anti-trichomoniasis drugs is currently observed and new drugs are being sought to combat this disease. Experiments were conducted to describe the activity of curcumin against *Trichomonas pochwylis*. CUR and quercetin loaded with hyaluronic acid niosomes were tested for their anti-Trichomonas activity, and their cytotoxic effects on a fibroblast cell line were also assessed. The results showed that quercetin and CUR at a concentration of 100 mg/mL after 24 h had anti-Trichomonas activity, while CUR at a concentration of 100 mg/mL at 3 h with 97% growth inhibition had a better effect than metronidazole (positive control). The obtained results also confirm that both curcumin and quercetin, even at the highest concentration (400 mg/mL), did not have a toxic effect on the fibroblast cell line [115]. 

Other researchers investigated the effects of CUR on proteolytic activity and hydrogenosomal metabolism in *Trichomonas pochwylis*. Moreover, the role of cucumin in pro-inflammatory responses induced in RAW264.7 phagocytic cells by parasitic proteinases on pro-inflammatory mediators such as nitric oxide (NO), tumor necrosis factor-α, interleukin-1beta (IL-1β); the heat shock chaperone was assessed against 70 (Hsp70) and the glucocorticoid receptor (mGR). It was observed that CUR can inhibit the growth of *T. pochwylis* trophozoites, increase the expression of pyruvate: ferredoxin oxidoreductase (PfoD), hydrogenosomal enzymes, and inhibit the proteolytic activity of parasitic proteinases. The researchers also demonstrated that CUR inhibited the production of NO and reduced the expression of pro-inflammatory mediators in macrophages. The conducted experiments confirmed the usefulness of CUR in controlling trichomoniasis and alleviating its associated immunopathogenic effects [116].

#### 2.6.4. Antiviral Activity

One type of skin viral infection is caused by the Herpes Simplex virus (HSV). The HSV herpes virus is divided into two types: HSV-1 and HSV-2. The first type (HSV-1) most often causes oral herpes, while the second type (HSV-2) is responsible for genital herpes. The mechanisms of curcumin’s antiviral action are related to the ability of curcumin to inhibit many cellular and molecular processes necessary for the expression, replication, and pathogenesis of viral genes [117]. The conducted research confirmed that curcumin inhibits the expression of the IE gene and hinders HSV-1 infection by using a mechanism independent of the activity of the p300/CBP histone acetyltransferase of transcription coactivator proteins [118]. Moreover, in vitro and in vivo analysis showed that curcumin inhibits HSV-2 replication via the NF-κB transcription factor [119]. The inhibition mechanism of HSV-1 and HSV-2 adsorption by curcumin was also observed in an in vitro experiment [120]. Curcumin’s anti-inflammatory activity has also been shown to be responsible for HSV-2 infectivity and replication [121]. 

Curcumin was analyzed for its local effect on the herpes simplex virus in the form of a proniosomal gel. Curcumin’s low solubility and penetration make it a suitable candidate for entry into proniosomes. Therefore, a topical CUR proniosome gel was developed and its in vitro and ex vivo activity against Herpes Simplex virus type 1 (HSV-1) was evaluated, as well as molecular docking studies of HSV-1 thymidine kinase proteins. Studies of the obtained CUR probiosomal gel showed a reduction in HSV-1 replication [122]. Other researchers investigated the effects of CUR -treated Herpes Simplex virus-1 (HSV-1) and Herpes Simplex virus-2 (HSV-2) virions on cultured Vero cells. Here, HSV-1 and HSV-2 virions were treated with curcumin at different maximum noncytotoxic concentrations. Experiments were performed using antiviral tests such as WST-1, plaque tests, adsorption, and penetration tests. When treating HSV-1 and HSV-2 viruses with CUR at a concentration of 30 μM, a reduction in the production of infectious HSV-1 and HSV-2 virions in cultured Vero cells was observed by interfering with the adsorption process [120].

One of the most common sexually transmitted viruses is Herpes Simplex virus type 2 (HSV-2), and it is a known risk factor for HIV infection in the female genital tract (FGT). The present study determined whether CUR nanoparticles delivered using various routes in vivo could minimize inflammation and prevent or reduce HSV-2 infection in FGT. Experiments were conducted in female mice that were pretreated with CUR nanoparticles using oral, intraperitoneal, and vaginal routes and then exposed vaginally to the tissue inflammation, stimulating CpG-oligodeoxynucleotide (ODN). The results confirmed that only intravaginal delivery of CUR nanoparticles reduces CpG-mediated inflammatory histopathology, and reduces the production of proinflammatory cytokines, interleukin (IL)-6, TNF-α, and monocyte chemoattractant protein-1 (MCP-1) in FGT. Ultimately, however, curcumin nanoparticles did not show antiviral activity or reduce tissue pathology when administered before intravaginal HSV-2 infection. Additionally, it was confirmed that intravaginal pretreatment with raw CUR or solid CUR dispersion forms showed increased survival and delayed pathology after HSV-2 infection [121]

### 2.7. Anti-Aging/Angiogenesis

The concept of angiogenesis can be defined as the formation of new blood vessels based on existing ones, under physiological conditions [123]. It is a short-lived and highly controlled process [124]. Despite a series of positive effects of this process, such as effects on growth, development, reproductive functions (maturation of the corpus luteum, regeneration of the endometrium and placenta) or recovery of wounds and fractures, it can promote the development of pathological conditions especially in the skin, such as skin cancer, atopic dermatitis, psoriasis, or complications of diabetes [123,124].

The inducers of angiogenesis mainly are growth factors such as basic fibroblast growth factor (bFGF), vascular endothelial growth factor (VEGF), or placental growth factor, but in the process, other factors—epidermal growth factor (EGF), transforming growth factors (TGF), fibroblast growth factors (FGF), angiopoietin-1 and 2, and matrix metalloproteinases (MMPs)—are active [125].

The effects of curcumin on angiogenesis are dual: CUR has shown the ability to stimulate angiogenesis, e.g., in wound healing [126] and ischemic skin conditions [127], but on the other hand, it can inhibit angiogenesis in cancerous conditions [125,128], psoriasis [129], and endometriosis [130].

Angiogenesis plays a relevant role in the wound healing process, and its induction is especially important in diabetic wounds, where levels of major angiogenic factors are decreased. CUR stimulates neovascularization in the wound area and significantly increases microvessel density, as observed in diabetic rats treated with curcumin. Increased expression of VEGF mRNA and TGF-β was also observed in the treated group. VEGF stimulates the formation of new blood vessels by encouraging the growth of endothelial cells and inhibiting their apoptosis. In addition, application of CUR caused upregulated expression of HIF-1α (hypoxia-inducible factor 1α), SDF-1α (stromal cell-derived factor 1 α), and HO-1 (heme oxygenase-1)—the factors that indirectly stimulate angiogenesis [126]. Moreover, the level of proangiogenic factors was found to be lower in the case of wounds caused by burns—a decrease in VEGF and α-SMA (alpha-smooth muscle actin) was detected. Following curcumin treatment in a burn wound model in rats, a significant increase in VEGF and α-SMA was observed, with a higher increase noted in the group treated with nano-curcumin (curcumin in nanoparticle form) [131].

Beyond the aforementioned, CUR contributes to the activation of angiogenesis in ischemic areas of the skin, in this case, by stimulating the reconstruction of the microvascular network, as shown in a mouse study. Moreover, in addition to the upregulation of VEGF expression, a significant increase in neovascularization-related proteins—MMP-9 and Cadherin 5—was detected [127].

Despite its pro-angiogenic potential, CUR has been found to have anti-angiogenic properties—especially, when angiogenesis is aggravated. In mice with induced psoriasis, a decrease in the expression of two biomarkers of neovascularization—VEGF and PECAM (Platelet Endothelial Cell Adhesion Molecule)—was noted. Furthermore, CUR treatment returned the reduced length and density of vessels to a normal level, indicating inhibition of angiogenesis, which is increased in psoriasis [129].

The anti-angiogenic effect of CUR in particular is evident in tumors. Curcumin has shown the ability to reduce levels of VEGF, a factor that plays a key role in the pathological angiogenesis that occurs in tumorigenesis [125]. Following the application of CUR, a decrease in VEGF was observed in the Xenograft model of melanoma in mice and also in HUVECs (human umbilical vein endothelial cells) [128], the Xenograft model of breast cancer [125], and endometriosis [130]. In addition to VEGF, CUR also inhibits the activity of PDE2 and PDE4 in the Xenograft melanoma model (PDE2 and PDE4 are also involved in angiogenesis) [128]. Furthermore, curcumin decreases the level of other factors that are expressed during tumor neovascularization such as bFGF and MMPs [125].

In all organisms, over time, aging processes are induced involving the inhibition of growth processes in favor of the dominance of atrophic processes. These are dynamic changes, the rate of occurrence of which depends on many factors, both genetic and environmental. The aging process is multi-level, starting with a single cell, through to tissues, and ending with organs. The most favorable model of aging is the cellular model, which helps accurately study the process and its stimulation using various factors. Currently, there are several theories of skin aging. These include gene theory, Hayflick theory, mitochondrial theory, membrane theory, and protein disorder theory. Changes that occur in the body with aging include a decrease in skin elasticity and tone, irregular color, and reduced work of the sebaceous glands, which contributes to increased water loss and disruption of the skin’s water–lipid mantle [132]. 

Despite the fact that these changes are not considered to determine serious health problems they mainly became an aesthetic issue. Modifications to the skin’s condition, which are related to the aging process, come from the notable decrease in the quantity of fibroblasts—cells crucial for collagen synthesis. Insufficient collagen levels in the skin result in its laxity and sagging. Moreover, the penetration of UV rays into the dermis stimulates the production of matrix metalloproteinases and free radicals. These compounds contribute to DNA mutations and instigate oxidative stress within the skin. This promotes the degradation of collagen, ultimately resulting in the formation of wrinkles and fine lines. 

Curcumin, as an antioxidant, has potential to prevent these changes and delay the aging process [133]. Additionally, CUR can also retard the aging process through inhibiting NFκB, reducing lipofuscin accumulation or enhancing the expression of the β1-integrin gene. β1-integrin is a cell surface receptor which accelerates wound healing, and increases the production of collagen and granulation tissue formation. It also induces fibroblasts differentiation into myofibroblasts [134,135].

Curcumin, as a natural hydrophobic dissolves easily in substances like dimethyl sulfoxide, acetone, ethanol, and oils [136]. However, it does not demonstrate good permeability through skin [137]. Similarly to other compounds that are delivered epidermally, CUR also must overcome obstacles such as the skin’s stratum corneum which results in decreased transdermal drug efficacy. Therefore, researchers are trying to find the perfect way of CUR formulations application so that the compound crosses the stratum corneum and reaches the dermis more efficiently [134]. 

To improve penetration of the curcumin through the skin, various nanosystems can be applied, for example liposomes, polymeric micelles, nanoemulgels, or carbon nanotubes. A study from 2024 shows that newly developed nanoemulgels perform better than existing formulations available in the market. Their efficacy and therapeutic potential are due to properties such as controlled drug release profiles and adequate ex vivo skin drug permeation and/or retention that ensure high levels of anti-aging qualities of curcumin formulation [137]. An intriguing way to topically apply CUR would be to use a face mask. In this mask, anti-oxidant properties of CUR are conserved and the bioactive compound is effectively delivered through the skin. This allows to conclude that curcumin face masks can be used to provide anti-aging benefits [138]. According to many recently published papers reviewed above, novel dosage forms, especially usage of nanotechnology in order to develop advanced delivery systems is the most promising approach in the topical administration of curcumin. Various nanoparticles have been obtained and investigated, although it seems that lipid nanoparticles may have the highest potential in improvement of curcumin bioavailability in the treatment of numerous different skin conditions/diseases. Nevertheless, promising effects have been reported for modified silica carriers as well as β-cyclodextrins in the form of smartPearls^®^ and nanosponges, respectively [41,44]. In Figure 4, selected novel nanoparticles that may be appropriate in topical curcumin delivery are presented.

## 3. Conclusions and Perspectives

Turmeric, which is the main source of curcumin, is known for its health-promoting properties since the ancient times. It is a plant that is especially popular among the inhabitants of India and South East Asian nations, and is grown in countries with a hot, subtropical climate. There is an increasing number of studies that state that curcumin can modulate phenomena associated with inflammatory, infectious, and proliferative skin diseases. The plant is widely available in most countries worldwide. It is applied for health-promoting purposes due to its multidirectional action, of which its antioxidant activity is a very important issue. The effect of curcumin on skin diseases is a noteworthy and practical issue; however, many studies require the aspect of increasing the absorption of this compound through the skin, as well as understanding the effect of polyphenols on many signaling cascades in the body. 

Curcumin modulates multiple molecular targets, including cytokines, chemokines, signaling proteins, cell cycle proteins, enzymes, receptors, and cell surface adhesion molecules. However, the health benefits of orally administered curcumin are often limited due to its low intestinal absorption, poor solubility, fast metabolism, and rapid elimination. Although the majority of ingested curcumin is excreted in the feces without being metabolized, a small portion is converted into water-soluble metabolites, such as sulfates and glucuronides. Recent advancements in medicinal technology have led to the development of several innovative curcumin preparations, which have significantly improved the bioavailability and safety of oral consumption. The first-generation preparation, using second-generation adjuvants and up to third-generation polysorbates, containing only natural materials, demonstrated significantly greater absorption capacity and cellular uptake, as well as improved safety, not only in patients but also in healthy individuals. This provides evidence for the potential therapeutic applications of these preparations.

CUR is a unique molecule with many applications and methods of administration, including oral and topical. However, topical administration of curcumin presents also several challenges due to its high lipophilicity and degradation, which limit its pharmacological effectiveness. It has high therapeutic potential, but its effectiveness is limited. To enhance the skin penetration of curcumin, various nanosystems can be employed, such as liposomes, nanoemulgels, polymer micelles, or carbon nanotubes. Lipid-based nanoparticles are particularly attractive due to their biocompatibility and similarity to the skin in terms of structure and composition. Encapsulation of curcumin in lipid-based nanoparticles may offer an effective solution to overcome the challenges associated with curcumin delivery, stability, solubility, permeation, and efficacy.

These nanoparticles can effectively trap curcumin and provide controlled release. Animal trials and studies conducted ex vivo, on the cell line, and in vitro have demonstrated the potential of lipid nanoparticles to deliver curcumin to the skin layers. However, there is still a research gap in the area of absorption, bioavailability, and penetration of curcumin and its new forms of administration. For this reason, future research must be directed towards overcoming the limitations of CUR described above. Further clinical trials are required to investigate the safety and effectiveness of curcumin preparations in large and diverse patient populations with various forms and phases of skin diseases.

## Figures and Tables

**Figure 1 ijms-25-03617-f001:**
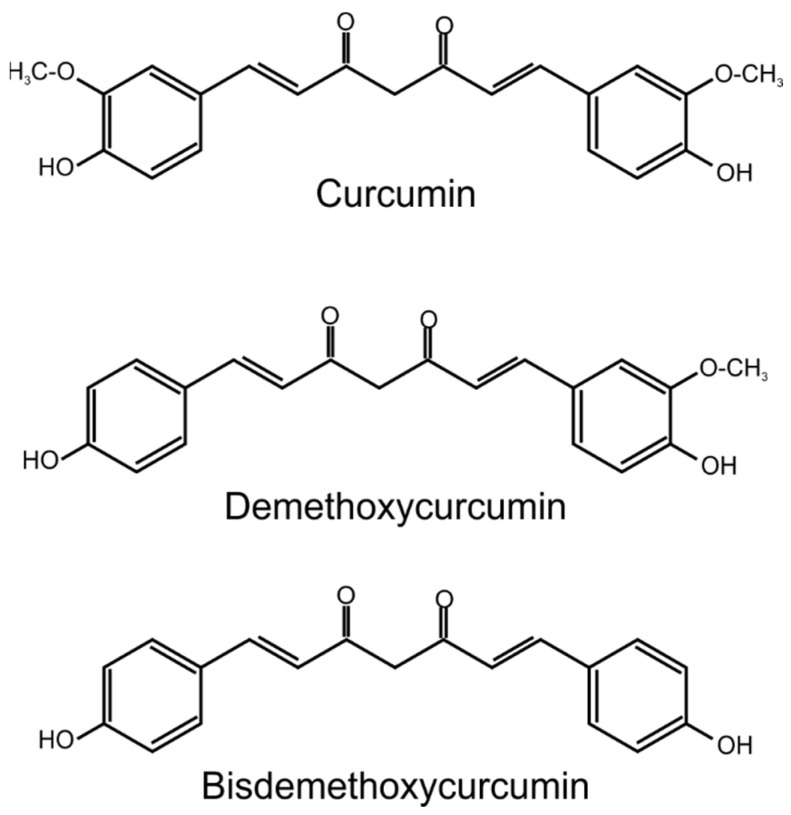
Chemical structure of curcuminoids.

**Figure 2 ijms-25-03617-f002:**
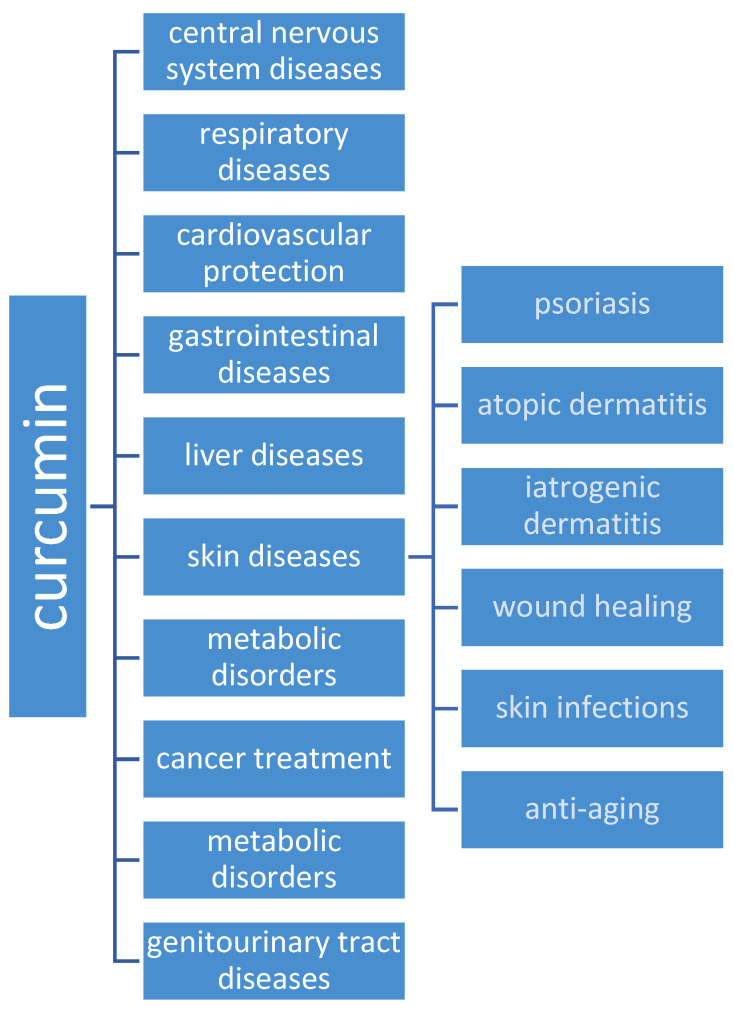
Directions for the health-promoting effects of curcumin.

**Figure 3 ijms-25-03617-f003:**
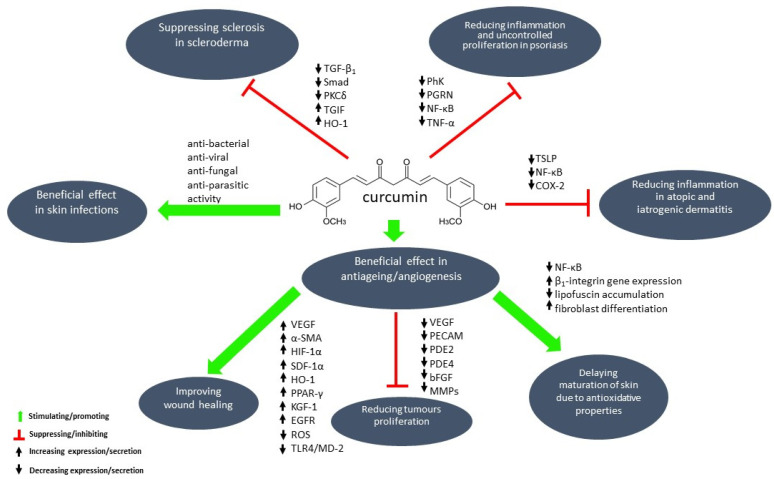
Molecular mechanisms of curcumin mode-of-action in selected skin diseases. Abbreviations: PhK—phoshphorylase kinase, PGRN—progranulin, NF-κB—nuclear factor kappa-light-chain-enhancer of activated B cells, TNF-α—tumor necrosis factor alpha, TGF-β_1_—transforming growth factor-beta 1, Smad—suppressor of mothers against decapentaplegic protein, TGIF—TGF-β_1_ induced factor, PKC—protein kinase C, HO-1—heme oxygenase-1, TSLP—thymic stromal lymohopoietin, COX-2—cyclooxygenase-2, PPAR-γ—peroxisome profiferator-activated receptor gamma, TLR4/MD-2—toll-like receptor 4/myeloid differentiation factor 2, ROS—reactive oxygen species, HIF-1α—hypoxia-inducible factor 1 alpha, SDF-1α—stromal cell-derived factor 1 alpha, α-SMA—alpha-smooth muscle actin, VEGF—vascular endothelial growth factor, PDE2, PDE4—phosphodiesterase type 2 and type 4, bFGF—basic fibroblast growth factor, PECAM—platelet Endothelial Cell Adhesion Molecule, MMPs—matrix metalloproteinases, KGF-1—keratinocyte growth factor-1, EGFR—epidermal growth factor receptor.

**Figure 4 ijms-25-03617-f004:**
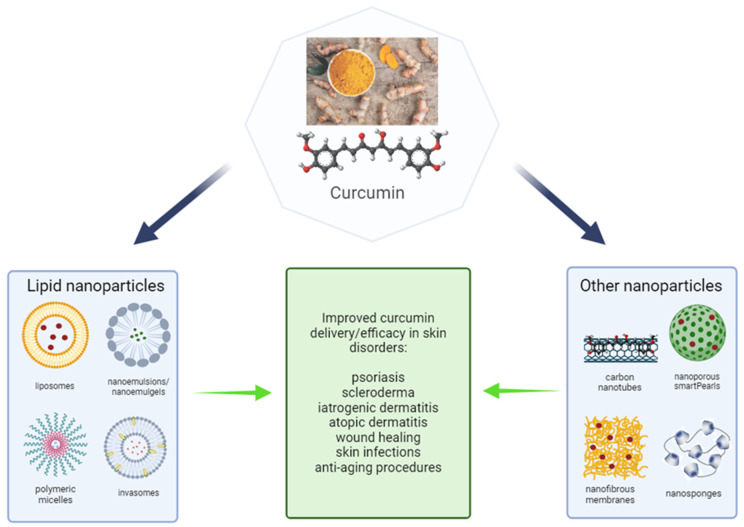
Selected nanoparticles with potential in the curcumin delivery systems for topical administration.

## Data Availability

The data presented in this study are available on request from the corresponding author.

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
