# Peer review of "Potential of Curcumin in the Management of Skin Diseases"

_ijms, 2024, doi:10.3390/ijms25073617_

Round 1

Reviewer 1 Report

Comments and Suggestions for Authors

The paper addresses a compelling topic and is well-written. However, I have several points and recommendations that warrant consideration:

  1. 1. I recommend the authors highlight details about the databases used for data collection/extraction (e.g., Web of Science, Scopus, Google Scholar) and the keywords employed during the literature search, alongside the period of studies included in the review. This ensures comprehensive coverage of recent and relevant studies, preferably in the introduction section.

  2. 2. I suggest augmenting the paper with additional information regarding curcumin and its promising effects against herpesviruses, which are pathogens known to induce skin diseases, oral herpes, and genital herpes sores. The authors can draw from the reference (DOI: 10.3390/microorganisms9020292) to extract relevant insights.

  3. 3. I recommend the authors create figures illustrating the molecular mechanisms induced by curcumin against the reviewed skin diseases and their pathogenic agents. Many readers find visual aids more accessible than textual descriptions or tables.

  4. 4. Please address the challenges and future directions associated with utilizing curcumin as a promising therapeutic agent in managing skin diseases.

  5. 5. Lastly, I recommend a thorough proofreading of the full text to identify and rectify any typographical errors. Also, ensure that all scientific names are properly italicized for adherence to scientific writing conventions.

Comments on the Quality of English Language

The English usage requires further refinement.

Author Response

The authors would like to thank the Reviewer for valuable comments which have helped to improve the quality of the manuscript. We hope that the revisions in the manuscript and our accompanying responses will be sufficient to make our manuscript suitable for publication.

The paper addresses a compelling topic and is well-written. However, I have several points and recommendations that warrant consideration:

  1. I recommend the authors highlight details about the databases used for data collection/extraction (e.g., Web of Science, Scopus, Google Scholar) and the keywords employed during the literature search, alongside the period of studies included in the review. This ensures comprehensive coverage of recent and relevant studies, preferably in the introduction section.

As the literature search engine, the browsers in the Scopus, PubMed, Web of Science databases and ClinicalTrials.gov register were used. The following inquiries were used: “skin diseases and curcumin”, “skin disorders and curcumin”, “psoriasis and curcumin”, “scleroderma and curcumin”, “dermatitis and curcumin”, „wound healing and curcumin”, ”skin infections and curcumin”, „anti‑aging and curcumin”, „angiogenesis and curcumin”, „curcumin and ‘in vitro’ studies”, „curcumin and ‘clinical trials’”, “curcumin and ‘animal models’”. Documents published from 1980 to January 2024 were included. In order to qualify work for the review the following exclusion criteria were applied: works do not contain original data e.g. reviews, comments, works which have not been independently peer-reviewed e.g. conference papers, letters to editor, pre-prints etc., works in other than English language, papers published before 1980 year.

These information has been added to the introduction.

  1. I suggest augmenting the paper with additional information regarding curcumin and its promising effects against herpesviruses, which are pathogens known to induce skin diseases, oral herpes, and genital herpes sores. The authors can draw from the reference (DOI: 10.3390/microorganisms9020292) to extract relevant insights. 

Thank you for valuable comment. The text has been supplemented with information regarding curcumin and its promising effects against herpesviruses.

  1. I recommend the authors create figures illustrating the molecular mechanisms induced by curcumin against the reviewed skin diseases and their pathogenic agents. Many readers find visual aids more accessible than textual descriptions or tables.

Thank you for comment. The Figure 3 illustrating molecular mechanisms of curcumin action in reviewed skin conditions has been implemented in the manuscript.

  1. Please address the challenges and future directions associated with utilizing curcumin as a promising therapeutic agent in managing skin diseases.

As recommended by the Reviewer, the Conclusion section identifies a research gap, challenges and future research opportunities regarding the administration of curcumin in skin diseases.

  1. Lastly, I recommend a thorough proofreading of the full text to identify and rectify any typographical errors. Also, ensure that all scientific names are properly italicized for adherence to scientific writing conventions.

Thank you for comment. The manuscript has been improved towards grammar and stylistics by native speaker Jack Stanley Dunster from Canada (language editor of Current Issues in Pharmacy and Medical Sciences).

Reviewer 2 Report

Comments and Suggestions for Authors

Dear Authors,

Curcumin and resveratrol are the wonder molecules that puzzle researchers with their diverse biological properties. The use of curcumin to manage skin disorders is widespread. Curcumin is an essential component of many cosmetic products. Hence, the submitted work is a good read. Before it gets accepted for publication the following changes/modifications are suggested to make the manuscript more interesting and self-explanatory,

1. Kindly replace the Rhizome figure of curcumin with a better one. A rhizome is not visible in the Graphical abstract.

2.  Kindly include the structure of bisdemethoxycurcumin, demethoxycurcumin, and curcumin and assign the figure number (Line 45).

3. Lines 45 and 47 are repeated. one can be deleted from the manuscript.

4. It's better to delete the word systemic from line 74. Just "Bioavailability" is enough.

5. Modify lines 75-77.  

6. DMSO dissolve most of the compounds. So, better delete it from line 79.

7. Authors have explained multiple skin diseases and disorders in the manuscript along with the role of curcumin. However, no graphical or pictorial representation is included for the same. It will make the manuscript more informative if images of psoriasis, scleroderma, atopic dermatitis and so on are included in the manuscript with proper reference.

8. Please include the pictorial representation of different formulations of curcumin with proper reference.

9. Kindly identify the research gap and future possible research in this field under the conclusion section. 

10. Kindly write all references uniformly. Few references have an Abbreviation for the Title name while few have their full name e.g., Reference numbers 129, 124, 120, 117 etc. 

11. Revise the manuscript thoroughly and check for the repeated words.

Author Response

The authors would like to thank the Reviewer for valuable comments which have helped to improve the quality of the manuscript. We hope that the revisions in the manuscript and our accompanying responses will be sufficient to make our manuscript suitable for publication.

Curcumin and resveratrol are the wonder molecules that puzzle researchers with their diverse biological properties. The use of curcumin to manage skin disorders is widespread. Curcumin is an essential component of many cosmetic products. Hence, the submitted work is a good read. Before it gets accepted for publication the following changes/modifications are suggested to make the manuscript more interesting and self-explanatory,

  1. Kindly replace the Rhizome figure of curcumin with a better one. A rhizome is not visible in the Graphical abstract.

Thank you for comment. Graphical abstract has been corrected as suggested.

  1. Kindly include the structure of bisdemethoxycurcumin, demethoxycurcumin, and curcumin and assign the figure number (Line 45).

Thank you for comment. The structure of bisdemethoxycurcumin, demethoxycurcumin, and curcumin are shown in Figure 1.

  1. Lines 45 and 47 are repeated. one can be deleted from the manuscript.

Thank you for comment. Repeated phrase has been removed.

  1. It's better to delete the word systemic from line 74. Just "Bioavailability" is enough.

Thank you for comment. The word „systemic” has been removed.

  1. Modify lines 75-77.  

Thank you for comment. Lines 75-77 have been corrected.

  1. DMSO dissolve most of the compounds. So, better delete it from line 79.

Thank you for comment, unnecessary phrase has been removed.

  1. Authors have explained multiple skin diseases and disorders in the manuscript along with the role of curcumin. However, no graphical or pictorial representation is included for the same. It will make the manuscript more informative if images of psoriasis, scleroderma, atopic dermatitis and so on are included in the manuscript with proper reference.

Our subject of interest is the biological activity of curcumin and other natural compounds, mainly from the polyphenols group. Unfortunately, we are not clinicians and do not have a photos showing the individual diseases mentioned in the text, and we do not want to obtain photos from the Internet (of course, this involves obtaining the rights from the authors) for fear that dermatological diseases will be poorly depicted there.

  1. Please include the pictorial representation of different formulations of curcumin with proper reference.

Thank you for comment. The Figure 4 illustrating selected novel formulations of curcumin has been implemented in the manuscript as well as proper explanation in the text.

  1. Kindly identify the research gap and future possible research in this field under the conclusion section. 

As recommended by the Reviewer, the Conclusion section identifies a research gap, challenges and future research opportunities regarding the administration of curcumin in skin diseases.

  1. Kindly write all references uniformly. Few references have an Abbreviation for the Title name while few have their full name e.g., Reference numbers 129, 124, 120, 117 etc. 

References have been corrected.

  1. Revise the manuscript thoroughly and check for the repeated words.

Thank you for comment. The manuscript has been improved towards grammar and stylistics by native speaker Jack Stanley Dunster from Canada (language editor of Current Issues in Pharmacy and Medical Sciences).

Round 2

Reviewer 1 Report

Comments and Suggestions for Authors

The manuscript has been sufficiently improved.